

# Assessing the tropical Atlantic biogeochemical processes in the Norwegian Earth System Models

Shunya Koseki[1], Lander R. Crespo[1], Jerry Tjiputra[2], Filippa Fransner[1], Noel S. Keenlyside[1,3] David Rivas[1,4]

[1]Geophysical Institute, University of Bergen / Bjerknes Centre for Climate Research, Bergen, 5007, Norway
[2]NORCE Norwegian Research Centre / Bjerknes Centre for Climate Research, Bergen, 5838, Norway
[3]Nansen Environment and Remote Sensing Centre/ Bjerknes Centre for Climate Research, Bergen, 5007, Norway
[4]Centro de Investigación Científico y de Educación Superior de Ensenada, Ensenada, 22860, Mexico

*Correspondence to*: Shunya Koseki (Shunya.Koseki@uib.no)

**Abstract.** State-of-the-art Earth system models exhibit large biases in their representation of the tropical Atlantic hydrography, with potential large impacts on both climate and ocean biogeochemistry projections. This study investigates how biases in model physics influences marine biogeochemical processes in the tropical Atlantic using the Norwegian Earth System Model (NorESM). We assess four different configurations of NorESM: NorESM1 is taken as benchmark

(NorESM1-CTL) that we compare against the simulations with (1) a physical bias correction and against (2 and 3) two configurations of the latest version of NorESM with improved physical and biogeochemical parameterizations with low and intermediate atmospheric resolutions, respectively. With respect to NorESM1-CTL, the annual-mean sea surface temperature (SST) bias is reduced largely in the first and comparably third simulations in the equatorial and southeast Atlantic. In addition, the SST seasonal cycle is improved in all three simulations, resulting in more realistic development of the Atlantic

Cold Tongue in terms of location and timing. Corresponding to the cold tongue seasonal cycle, the marine primary production in the equatorial Atlantic is also improved and in particular, the Atlantic summer bloom is well represented during June to September in all three simulations. The more realistic summer bloom can be related to the well-represented shallow thermocline and associated nitrate supply from the subsurface ocean at the equator. The climatological intense outgassing of sea-air $CO_2$ flux in the western basin is also improved in all three simulations. Improvements in the

climatology mean state also lead to better representation of primary production and sea-air CO2 interannual variability associated with the Atlantic Niño and Niña events. We stress that physical process and its improvement are responsible for modeling the marine biogeochemical process as the first simulations, where only climatological surface ocean dynamics are corrected, provides the better improvements of marine biogeochemical processes.



## 1 Introduction

The tropical Atlantic Ocean is a region with intense biogeochemical cycling and productive ecosystems resulting in a hotspot for large fisheries (Gregg et al., 2003; Menard et al., 2000). In particular, the characteristics of the marine ecosystems in the tropical Atlantic are manifested by the high marine biological production along the west African coast associated with the Canary and the Benguela upwelling systems (Hutchings et al., 2009; Santos et al., 2007; Shannon et al., 2004; Vazquez et al., 2022). Another key driver of the marine ecosystem in the tropical Atlantic is riverine flux from the great rivers like the Congo and Amazon Rivers (Araujo et al., 2014; Bouillon et al., 2012; Demaster and Pope, 1996; Moreira-Turcq et al., 2003; Vieira et al., 2020). The coastal upwelling and riverine fluxes are important sources of nutrients such as nitrate ($NO_3^-$), phosphate ($PO_4^{3-}$), and silicate ($SiO_2$) for phytoplankton (Gao et al., 2023). Apart from the coastal areas, high marine production is also observed in the central to eastern basin of the equatorial Atlantic where the Atlantic Cold Tongue (ACT, Crespo et al., 2019; Hummels et al., 2013; Okumura and Xie, 2006; Tokinaga and Xie, 2011), associated with cold sea surface temperature (SST), develops during boreal summer (June-July-August). Here, a seasonal high production is fueled by the equatorial upwelling that supplies nutrient-rich seawater from the subsurface ocean (Chenillat et al., 2021; Kawase and Sarmiento, 1985; Perez et al., 2005). In addition to this predominant seasonal variation, the primary production in the equatorial Atlantic has a strong inter-annual variability associated with the Atlantic Niño and Niña (Crespo et al., 2022; Keenlyside and Latif, 2007; Prigent et al., 2020) that has its peak during boreal summer (Chenillat et al., 2021). The Atlantic Niño and Niña are, in general, induced by modifications in the equatorial upwelling and thermocline zonal gradient via the Bjerknes Feedback (Bjerknes, 1969; Crespo et al., 2022; Keenlyside and Latif, 2007; Prigent et al., 2020) while other possible mechanisms are also discussed such as thermodynamical driver and warm water advection from subtropical (Nnamchi et al., 2021; Nnamchi et al., 2015; Richter et al., 2013). Chenillat et al. (2021) showed that the upwelling changes associated with such Atlantic dynamical variability mode is predominantly responsible for the interannual variability in the equatorial Atlantic summer high production.

In addition to the high productivity, the tropical Atlantic Ocean plays an important role in the global carbon cycle (Takahashi et al., 2002). Model projections indicate that the tropical Atlantic is a key convergence zone for anthropogenic carbon in the future (Tjiputra et al., 2010), with rapid and long-term climate change imprints, such as warming, ocean acidification, and oxygen changes in the future (Bertini and Tjiputra, 2022; Tjiputra, 2023). The sea-air carbon dioxide ($CO_2$) flux in the tropical Atlantic Ocean is predominantly outgassing, making it the second largest $CO_2$ outgassing system in the global ocean (Sarmiento, 2006). This large $CO_2$ outgassing is mainly attributed to rich dissolved inorganic carbon that is supplied from subsurface ocean by the equatorial upwelling (Koseki et al., 2023) and enhances the surface partial pressure of $CO_2$ ($pCO_2$). In addition to dissolved inorganic carbon, $pCO_2$ is a function of several oceanic physical-chemical properties like SST, sea surface salinity (SSS), and total alkalinity (Sarmiento and Gruber, 2006). Lefevre et al. (2013) suggested that SST and SSS positive anomalies in the northern tropical Atlantic enhance the outgassing of $CO_2$ flux during February to



May. More recently, Koseki et al. (2023) showed a unique pattern and mechanism of $CO_2$ flux anomalies associated with the Atlantic Niño and Niña, which is distinct from that in the tropical Pacific (Ayar et al., 2022).

With the rapid development of computational technologies and resources, marine biogeochemical models are now standard components of Earth system models (ESMs), which have become key tools to investigate the global carbon cycle, marine physical-biogeochemical interaction and their feedbacks on the global and regional climate (Doney, 1999; Ilyina et al., 2013; Kriest and Oschlies, 2015; Seferian et al., 2020; Sein et al., 2015). They are also widely used to produce near-term predictions of the interannual to decadal evolution of the marine biogeochemistry (Fransner et al., 2020; Seferian et al., 2018; Seferian et al., 2019). These prediction models have added important evidence that ocean physics plays a major role in

shaping marine biogeochemical processes. For example, Ramirez-Romero et al. (2020), using four different coupled physical-biogeochemical model configurations, suggested that the intensity, timing and vertical location of deep chlorophyll maximum are very sensitive to the ocean stratification period and intensity. Fransner et al. (2020) showed that physical processes play a crucial role in controlling the nutrients and primary production variability and consequently the predictability of key biogeochemical processes such as $CO_2$ fluxes. It had been demonstrated that biases in physical

dynamics can bring about large uncertainty in future projections of ocean carbon sink (Bourgeois et al., 2022; Goris et al., 2023; Goris et al., 2018). Therefore, to improve the fidelity of future projections of ocean carbon cycle at regional scales, it is very important to understand the physical-biogeochemical interactions and verify how properly such interaction is simulated by the ESMs.

As a long-standing common issue, most of the advanced ESMs exhibit non-negligible systematic physical biases in
the representation of climate variables in the tropical Atlantic such as SST, precipitation, and other relevant atmospheric and oceanic fields (de la Vara et al., 2020; Koseki et al., 2018; Mohino et al., 2019; Voldoire et al., 2019), which can degrade predictability of climate variability  (Counillon et al., 2021). The origins of such systematic biases are diverse among the ESMs: imperfect parameterization of ocean mixed layer processes (Deppenmeier et al., 2020), coarse resolution of atmospheric and oceanic components (de la Vara et al., 2020; Harlass et al., 2018), intrinsic atmospheric bias of surface wind
(Koseki et al., 2018; Xu et al., 2014) and poor representation of subtropical atmospheric surface circulation (Cabos et al., 2017). The tropical Atlantic SST biases also exacerbate the climate variability and predictability (e.g., Counillon et al., 2021; Dippe et al., 2018; Prodhomme et al., 2019). While these physical and dynamical biases of the ESMs have been widely discussed during this decade, there are limited studies on their impact on the simulated marine biogeochemical processes in the tropical Atlantic.

Here, we assess the impact of physical and dynamical biases on the representation of biogeochemistry in the tropical Atlantic in one CMIP (Coupled Model Intercomparison Project) -class ESM, the Norwegian Earth System Model (NorESM). We evaluate theree simulations with (1) physical bias correction, (2) better parameterizations of atmosphere/ocean physical and marine biogeochemical processes, and (3) refinement of atmospheric model spatial resolution. Focusing on physical properties like SST and the thermocline, we investigate to what extent the biogeochemical
processes are improved in terms of climatology, seasonality, and inter-annual variability. This paper is structured as follows.



Section 2 gives the details of NorESM, its experimental settings, and the observational data used for verification. In Section 3, we show and discuss the results of NorESM simulations. Finally, this paper is summarized in Section 4.

## 2 Norwegian Earth System Model and Data

### 2.1 Model description

The first generation Norwegian Earth System Model (NorESM1, Bentsen et al., 2013), which contributes to the phase 5 of CMIP exercise (Taylor et al., 2012), consists of the Community Atmospheric Model version 4 (CAM4, Neale, 2010), the Miami Isopycnic Coordinate Model (MICOM; (Bleck et al., 1992), the Community Sea Ice Model (CICE4), the Community Land Surface Model (CLM4) and the Hamburg Ocean Carbon Cycle model (HAMOCC, Tjiputra et al., 2013). NorESM2 is the latest generation of NorESM with updates and tunings of physical and biogeochemical parameterization

((Seland et al., 2020; Tjiputra et al., 2020) and contributor to CMIP6 (Eyring et al., 2016). The atmospheric component is updated to CAM6-Nor with axial angular momentum conservation (Toniazzo et al., 2020) and parameterization for atmosphere-aerosol-radiation is employed. The ocean component of NorESM2 is replaced with the Bergen Layered Ocean Model (BLOM) that implements the updated parameterization of second-order closure scheme (Ilicak et al., 2008). HAMOCC is updated to iHAMOCC (Tjiputra et al., 2020). More details of NorESM2 description and broad scale evaluation

of its physics and ocean biogeochemistry are available in (Seland et al., 2020; Tjiputra et al., 2020).

### 2.2 Model configurations

With NorESM1 we performed a standard historical simulation. As a benchmark simulation, referred to as NorESM1-CTL, NorESM1 was initialized at 1980-01-15 from a historical spin-up starting at 1850-01-01 following

(Counillon et al., 2021). The initial conditions of HAMOCC we obtained from a historical run of (Tjiputra et al., 2013). NorESM1-CTL was integrated until the end of 2019. In the second model configuration, an anomaly coupling technique (Toniazzo and Koseki, 2018) was implemented into NorESM1 to reduce physical biases. In this methodology, the model's monthly climatologies of SST and surface wind were replaced by the observed ones during the model integration at every coupling step while the frequency of air-sea coupling was kept identical to NorESM1-CTL. The observed SST and surface

wind were obtained from HadISST and ERA-Interim (Dee et al., 2011) respectively for 1980-2000. This run is referred to as NorESM1-AC, and ocean carbon cycle is included as in NorESM1-CTL. Other details of NorESM1-CTL and NorESM1-AC (for example, spin-up duration, model performance, etc) can be found in Counillon et al. (2021). Due to the initial physical adjustments on the biogeochemistry, we considered the first 10 years of NorESM1-CTL and NorESM1-AC as adjustment period and were not analyzed in our study.

Two historical runs of NorESM2 (NorESM2-LM and NorESM2-MM) following the standard CMIP6 protocol were integrated from 1850 until 2014 and the data from 1990 to 2014 period are analyzed in this study. NorESM2-LM and NorESM2-MM differ in the spatial resolutions of the atmospheric model CAM6-Nor with a coarse resolution of 2.5°×1.9°



and an intermediate resolution of 1.5°×0.9°, respectively. The resolution of the ocean component is similar in all simulations of NorESM1 and NorESM2. On the other hand, the resolution of atmospheric components is equal for NorESM1 and NorESM2-LM. The simulations of NorESM1 and NorESM2 each have 5 and 3 ensemble members, respectively. These experimental settings are given in Fig. S1. To summarize, NorESM1-AC is a reference for physical bias correction and NorESM2-LM/MM are for improved physical and biogeochemical parametrizations in comparison with the benchmark simulation of NorESM1-CTL. We also aim to qualitatively assess the impacts of model refinement on simulation performance by comparing with NorESM1-CTL with NorESM2-LM and NorESM2-MM.

### 2.3 Observational data

We evaluate the NorESM simulations using observational datasets. The SST data is from Optimum Interpolated SST (OISST, Reynolds et al., 2007) from1990 to 2019. Three dimensional ocean data of temperature, nitrate and phosphate were taken from World Ocean Atlas 18 (WOA18, Locarnini et al. 2018; Garcia et al., 2018) climatological data. Monthly marine primary production was taken from MODIS (Moderate Resolution Imaging Spectroradiometer) satellite data from 2003 to 2019. The ocean surface $CO_2$ flux is from the global observation-based gridded data of Landschutzer et al. (2016) and Landschutzer et al. (2020) from 1990 to 2015.

## 3 Results

### 3.1 Climatology

First, we assess the SST bias in our four experiments (Fig. 1). NorESM1-CTL has a warm bias along the west African coast (Fig. 1a), which is a common bias in ESMs (Richter, 2015). In contrast, cold SST biases are detected in the subtropics. The causes of the SST bias in NorESM1 are predominantly erroneous wind stress and air-sea heat flux (Koseki et al., 2018). By implementing the anomaly coupling technique (NorESM1-AC), the tropical Atlantic SST biases are substantially alleviated (Fig. 1b, e). In particular, the warm bias of the Angola-Benguela Frontal Zone (ABFZ, 15°S to 17°S along the western African coast, e.g., (Koseki et al., 2019) is reduced by up to 5°C. NorESM2-LM also exhibits a considerably warm bias in the eastern tropical Atlantic while the subtropical cold biases are reduced at the south and even suppressed in the north (Fig. 1c). The improvement of the subtropical Atlantic is comparable with that of NorESM1-AC (Fig. 1e and f). The summer (June-July-August) SST bias is comparably alleviated between NorESM1-AC and NorESM2-LM (Fig. S2). In NorESM2-MM, the SST bias is reduced more than NorESM2-LM (Fig. 1d). The ABFZ warm bias in NorESM2-MM is improved by 3°C and the equatorial Atlantic by 2°C (Fig. 1g and Fig. S2). Comparison between NorESM2-LM and NorESM2-MM suggests that a horizontal refinement of the atmospheric model improves the climatic state of the surface ocean, consistent with (Harlass et al., 2018).




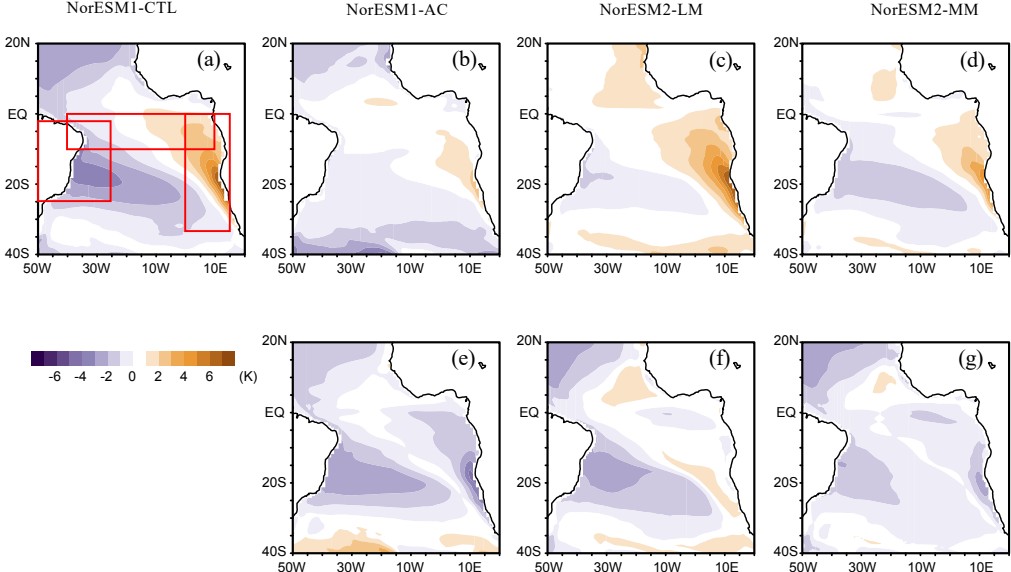

**Figure 1:** (a)-(d) Annual-mean climatological bias of sea surface temperature (SST) with respect to OISST data and (e)-(g) bias improvements of each simulation compared to NorESM1-CTL. In (e)-(h), the negative (positive) values indicate improvement (exacerbation) compared to NorESM1-CTL. The red boxes denote the area for averaging in Fig.2.

Figure 2 provides vertical sections of the observed and simulated ocean temperature around the south pan-tropical Atlantic Ocean. In the observation, a thick warm layer forms around the northeast Brazilian coast and western equatorial Atlantic while a thin warm layer penetrates from the eastern equatorial Atlantic to the ABFZ resulting in the east-west tilting thermocline depth along the equator (Fig. 2a). NorESM1-CTL fails to reproduce the east-west steep gradient of thermocline along the equator and the observed warm pool in the western Atlantic and northeastern Brazilian coast (Fig. 2b). The thick warm layer is homogeneously formed along this pan-tropical Atlantic sector and the ABFZ is pushed further southward. By applying the physical bias reduction (NorESM1-AC), the equatorial thermocline zonal-gradient bias is alleviated and the thick warm pool is generated more realistically than in NorESM1-CTL (Fig. 2c). The erroneous southward penetration of warm water along the African coast is suppressed, resulting in reduction of the warm SST bias in NorESM1-AC (Fig. 1b, c). While the zonal-tilting of the equatorial thermocline is well represented in NorESM2-LM, the warm pool is relatively shallower than NorESM1-AC in the western Atlantic and the ABFZ is pushed further southward comparable with NorESM1-CTL (Fig. 2d). In NorESM2-MM, the tilting thermocline is similarly well represented along the equator, and the location of the ABFZ are more realistic than NorESM2-LM. Compared to observation and NorESM1, NorESM2 tends to have warmer subsurface ocean (Fig. 2d and e).



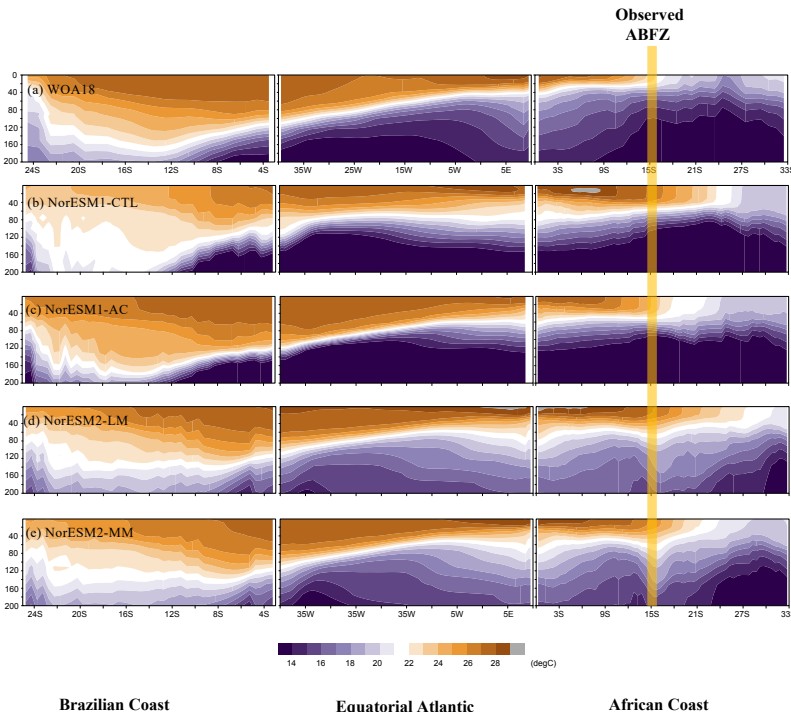

**Figure 2:** Depth sector of annual-mean climatology of ocean temperature along Brazilian coast, equatorial Atlantic, and African coast for observation and each NorESM simulation avearaged in the three boxes shown in Fig.1a. Yelllow line denotes the location of the Angola-Benguela Frontal Zone (ABFZ) in the observation.

## 3.2 Seasonality

Figure 3a-e illustrates temporal-longitude Hovmöller plots of SST in the equatorial Atlantic for observation and each model simulation. In the observations, the SST shows a clear seasonal cycle (Crespo et al., 2019; Ding et al., 2009) with the ACT developing in the boreal summer. NorESM1-CTL reproduces roughly the seasonal cycle of SST, but it fails to simulate the location and timing of the ACT: the ACT peak occurs more westward in the equator (30°W) and its peak is slightly later than in the observation (Fig. 3b). This discrepancy is consistent with the thick and zonally uniform warm layer along the entire equatorial Atlantic (Fig. 2b). Employment of the climatological bias correction leads to a more realistic development of the ACT, in particular, the location of the ACT is well represented (Fig. 3c; (Toniazzo and Koseki, 2018). Note that the anomaly coupling corrects directly the climatological surface wind forcing in the ocean model. In NorESM2 simulations, the SST seasonal cycle is also improved and NorESM2-MM has a stronger ACT with better timing during summer than NorESM2-LM (Fig. 3d and e).

Next, we investigate the simulation in surface biogeochemistry, which is tightly linked to physical dynamics and SST (e.g., Chenillat et al., 2021). Figure 3f-j shows the temporal-longitude Hovmöller plot of climatological primary production for observation and each simulation. In the observations, the primary production has a clear seasonal cycle with a



peak between 20°W and 0° in JJA (0.075 mol C m⁻² day⁻¹), which is consistent with the spatiotemporal development of the ACT (Fig. 3a, f). There is another less pronounced high productivity season during November to January in the equatorial

200    Atlantic (Fig. 3f). NorESM1-CTL simulates the summer bloom very poorly (Fig. 3g).

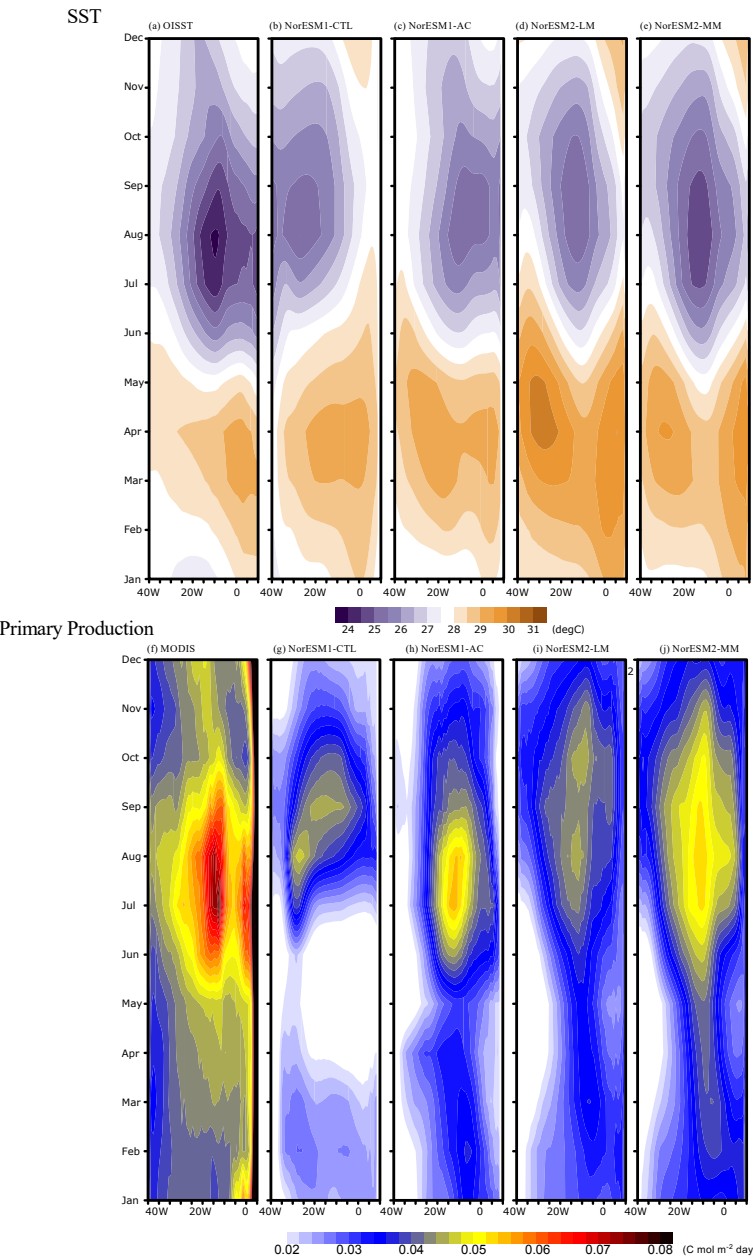

**Figure 3:** Climatological seasonal cycle of (upper row) SST and (lower row) primary production for observation and each simulation of NorESM along the equator (averaged 3S-3N). The observed primary production is obtained from MODIS satellite data. The modelled primary production is vertically integrated through the entire ocean layer.



205

The peak of the summer bloom is weaker, located more westward (30°W), and occurs later, in August and September, than in the observations. Apart from the summer bloom, there is another peak in February in the western basin and nearly no production in April to May. Interestingly, the climatological bias corrected simulation NorESM1-AC is able to reproduce the

210 observed timing and location of the summer bloom (Fig. 3h). The intensity of the summer bloom also increases (up to 0.055 mol C m$^{-2}$ day$^{-1}$) even though it is 27% lower than the observations. In the two NorESM2 simulations, the summer bloom tends to be better represented than in NorESM1-CTL (Fig. 3i and j). However, the summer bloom in NorESM2-LM is weak (approximately 0.043 mol C m$^{-2}$ day$^{-1}$) and there is a double-core peak in August and October. On the other hand, NorESM2-MM has a stronger summer bloom with a more realistic timing similar to NorESM1-AC. These differences in primary

215 production in the NorESM2 simulations can be attributed to the differences in the ACT development (Fig. 3d and e). All the NorESM simulations fail to reproduce the very high coastal production in the east, which will be discussed in the last paragraph of this subsection.

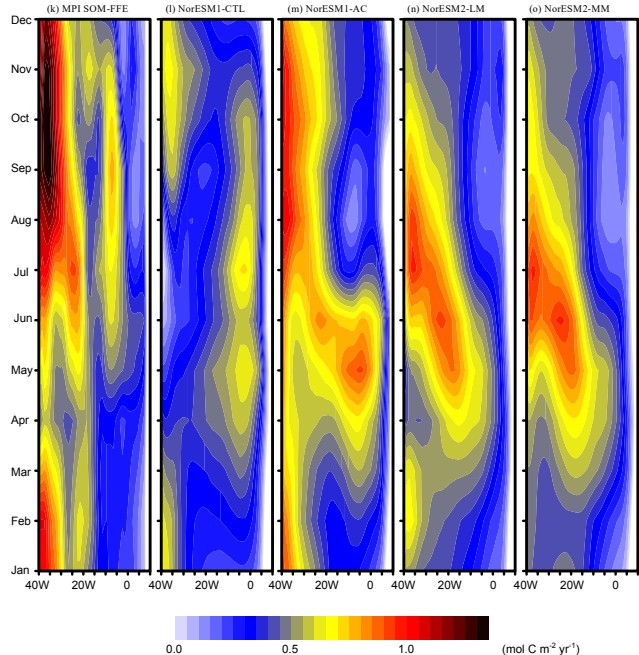

220  **Figure 3:** Continued. Climatological seasonal cycle of sea-air $CO_2$ flux. Positive value denotes upward.



The Hovmöller plot of sea-air $CO_2$ flux along the equator is given in Fig.3k-o. In the observations, the $CO_2$ flux has a clear seasonal cycle: particularly, maximum $CO_2$ flux outgassing during July to October in the western (40°-30°W) and eastern (10°W-0°) basins while the outgassing is modest in the central (20°W) basin (Fig. 3k). The late summer peak of the $CO_2$ flux in the central-eastern basin could be associated with the development of ACT that supplies the anomalously high dissolved inorganic carbon (DIC) water mass from the subsurface (Koseki et al., 2023). Contrastingly, in the western basin where such upwelling is weaker the outgassing may be related to the solubility of $CO_2$ gas. As Lefevre et al. (2013) and Koseki et al. (2023) suggest, the solubility of $CO_2$ gas (a function of temperature and salinity) is responsible for the inter-annual variability in $pCO_2$ and consequently sea-air $CO_2$ flux in the tropical Atlantic. In the western basin, the $CO_2$ outgassing is moderate in April when the precipitation is strongest (not shown) along the western equatorial Atlantic and in contrast, the timing of intense outgassing (August to October) is consistent with the period when the inter-tropical convergence zone (ITCZ) sits further northward from the equator.

NorESM1-CTL poorly reproduces the seasonal march of $CO_2$ distribution (Fig. 3l): the eastern outgassing shifts more eastward and it occurs one or two months earlier. In the western basin, the observed vigorous outgassing is not simulated well, except for some weak outgassing from September to March. In NorESM1-AC, the observed outgassing in the western basin is particularly well simulated from July to November although its magnitude is relatively modest (Fig. 3m). In the central to eastern basin, the early occurrence of intense outgassing remains. Similar to the primary production, improvement in the two NorESM2 simulations (Fig. 3n and o) relative to NorESM1-CTL is also evident for $CO_2$ flux. Nevertheless, the timing of the seasonal cycle in the eastern basin shifts considerably.

Compared to NorESM1-CTL, all other NorESM simulations improve the SST, primary production, and sea-air $CO_2$ flux seasonal cycle in a statistical way (Fig. 4). In particular, NorESM1-AC performs the best, followed by NorESM2-MM in reproducing the observed seasonal variations in SST and correspondingly sea-air $CO_2$ flux, and primary production (Fig. 4a). The pronounced improvements in the NorESM1-AC indicates that the atmospheric circulation is crucially responsible for representation of SST, PP and $CO_2$ flux in the tropical Atlantic. Indeed, the SST in this region is highly influenced by the wind inducing upwelling (e.g., Voldoire et al., 2019), which also supplies nutrients to the surface ocean that fuels PP. The improvement of sea-air $CO_2$ flux is almost identical between NorESM2-LM and NorESM2-MM. A scatter plot between SST and biogeochemical correlations clearly shows that the better simulation of SST seasonal cycle is important for simulating the seasonal cycle of biogeochemical processes (Fig. 4b).

Because the summer bloom in the tropical Atlantic is connected closely to the availability of nutrients (e.g., Radenac et al., 2020), here we assess the subsurface nutrient concentrations during JJA (Fig. 5). In the observations, nitrate ($NO_3^-$) and phosphate ($PO_4^{3-}$) have clear west-east tilting slopes associated with the thermocline during JJA (Fig. 5a, f, and k). According to Radenac et al. (2020), this nutrient supply to the euphotic zone is mainly driven by vertical advection associated with upwelling while vertical diffusion and meridional advection contribute to shape and spread the Atlantic summer bloom. As shown in Figs. 2b and 5b, the NorESM1-CTL fails to simulate the observed equatorial thermocline gradient. Corresponding to the flat thermocline, the upwelling of nitrate and phosphate is suppressed in the central to eastern



basin (Fig. 5g and l). In addition, the amount of nutrients is overestimated in the west (35°W-30°W) between 60 and 100 m depths. The westward-shifting and weaker summer bloom of production might be attributable to this nutrient supply bias in NorESM1-CTL. The alleviation of the thermocline bias by the climatological physical bias correction leads to a better

representation of the pumping of subsurface nutrients from the central to eastern basin (Fig. 5h and m). Similar improvement can be detected in NorESM2 simulations (Fig. 5i, j, n and o) resulting in a better seasonal cycle of the primary production, especially, the Atlantic summer bloom (Fig. 3i and j). In the two NorESM1 versions, the ocean subsurface is cooler and more abundant in nutrients than in NorESM2s, which could be associated with the difference in the ecosystem parameters, in addition to the ocean circulation, i.e., stronger Atlantic overturning circulation (Tjiputra et al., 2020) .

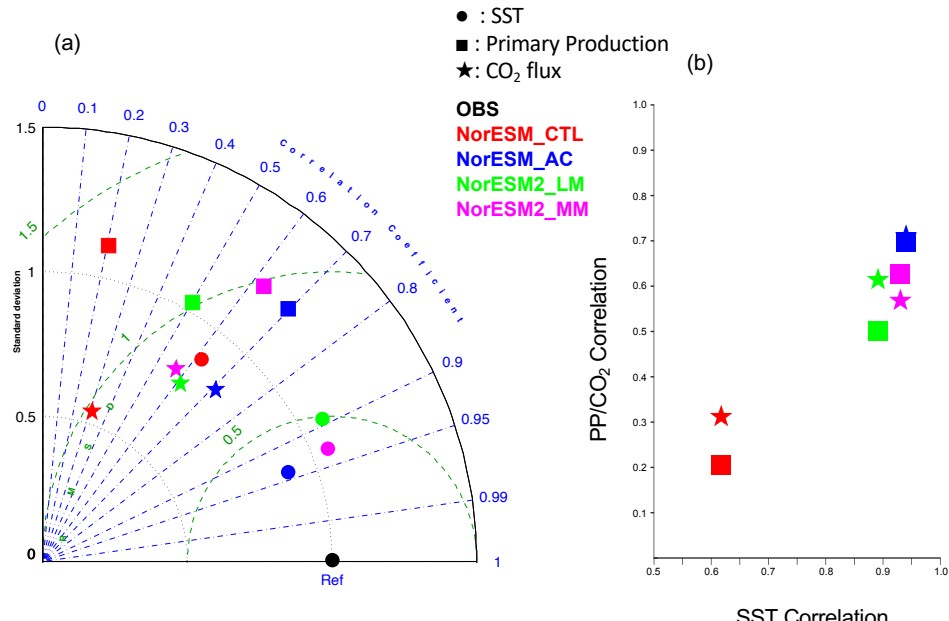

**Figure 4:** (a) Taylor diagram of the climatologicalseasonal cycle of SST (closed circle, primary production (closed square) and sea-air CO2 flux (star) with respect to observation of OISST, MODIS, and MPI SOM-FFE, respectively. Each NorESM simulation is distinguised by different color (NorESM1-CTL: red, NorESM1-AC:blue, NorESM2-LM: green, NorESM2-MM: magentha). (b) Scatter plot between SST correlation coefficient and PP/CO$_2$ flux. The convention of color and marker is same as (a). Note that the standard deviation is

normalized by that of observation and that the calculation of correlation and startd deviation do not include the data along the African coast.

Similar to the equatorial Atlantic, the climatologically-physical bias correction is beneficial for the coastal upwelling and nutrient supplies in the South Atlantic and western African coastal region where the marine biogeochemical

cycle and ecosystem are very intense (Figs. S3 and S4, e.g., Cury and Shannon, 2004; Shannon et al., 2004). NorESM2-MM simulates better coastal upwelling and nutrients than NorESM2-LM indicating that the horizontal refinement of the atmospheric component is also beneficial for the coastal upwelling. While the improved nutrient supply can be effective for the primary production in the Benguela upwelling region (between 15°S and 35°S) in NorESM1-AC (Fig. S4), the primary



production in the Benguela upwelling region in the two NorESM2 simulations is greatly reduced compared to NorESM1-
CTL. This might be caused by the parameter tuning in biological dynamics processes that suppress the anomalously excess
primary production here and in other oceanic regions (Tjiputra et al., 2020). In contrast, NorESM2 has slightly more primary
production in the equatorial coastal region (between 5°S and 10°S) than NorESM1 (Fig. S4). This can be attributed to the
riverine-originated nutrient input from the Congo River implemented in NorESM2 (Gao et al., 2023; Tjiputra et al., 2020).

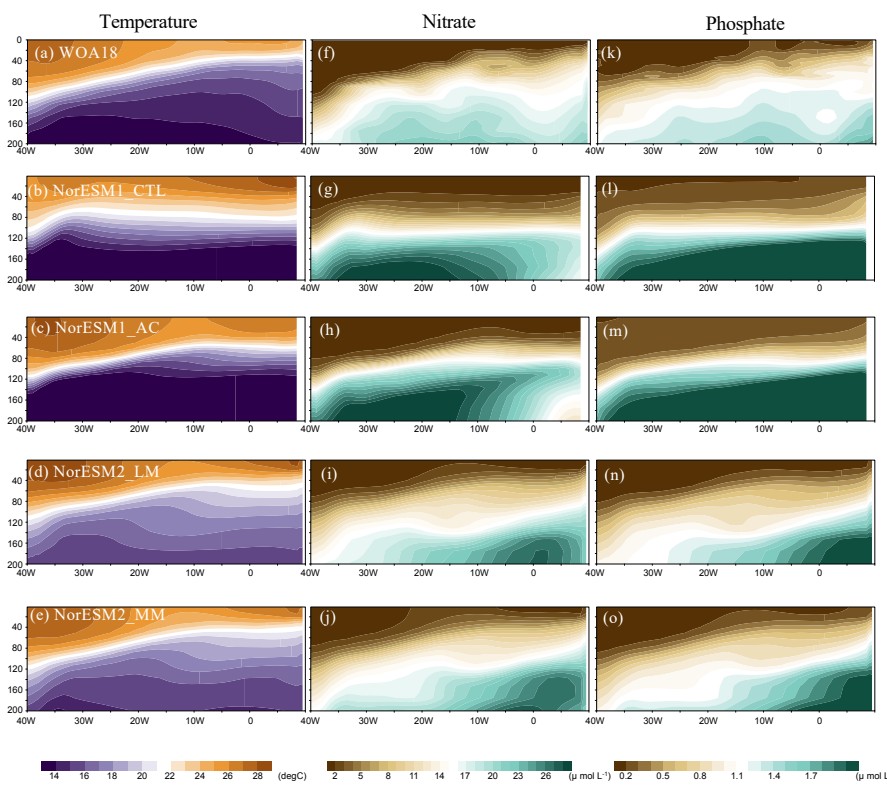


**Figure 5:** Depth-lonitudinial section of (left) temperature, (middle) nitrate, and (right) phosphate in JJA climatology for the observations
and each NorESM simulation averaged over 3°S and 3°N.

### 3.3 Interannual variability

290       One of the most pronounced climate variability patterns in the tropical Atlantic is the Atlantic Zonal Mode (AZM;
e.g., Keenlyside and Latif, 2007), referred as Atlantic Niño variability. As previous studies suggest (e.g., Counillon et al.,
2021; Dippe et al., 2018), the climatological biases adversely affect the simulation of SST variability in the tropical Atlantic.
In this section the Atlantic Niño variability and its impacts on the marine biogeochemical processes are assessed.

         Figure 6a-e illustrates the seasonality of SST inter-annual variability along the equator. In the observations, the peak
of variability associated with the Atlantic Niño and Niña events is found from June to July at around 20°W (e.g., Dippe et al.,



2018; Nnamchi et al., 2015). Apart from the summer, there is a secondary peak during November to December (e.g., Okumura and Xie, 2006). NorESM1-CTL, to some extent, is able to reproduce the observed seasonality of SST variability, however its summer peak is delayed by one month and the winter peak appears one-month earlier in November (Fig. 6b). During the autumn, the variability is unrealistically strong compared to the observations. In contrast, NorESM1-AC is
successful in simulating the summer and winter peaks with the right timing although the amplitude is weaker (Fig. 6c). Another study suggests that this improvement of variability is attributed to the improvement of the Bjerknes Feedback (e.g., Ding et al., 2015). While NorESM2-LM also reproduces the summer and winter peaks, this realization tends to overestimate

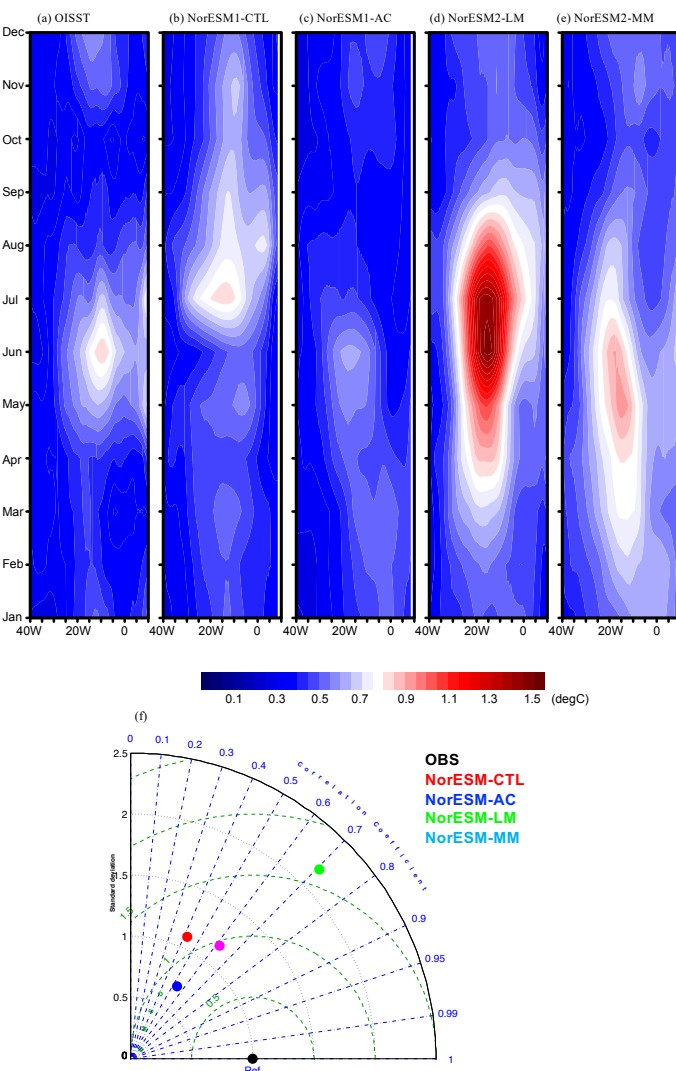

**Figure 6:** (a)-(e) same as in Fig. 3, but for SST standard deviation along the equator. (f) Same as in Fig. 4, but for the SST standard
deviation.





the inter-annual variability, particularly, in summer (Fig. 6d). NorESM2-MM is also able to improve the SST variability
having an overestimated summer peak amplitude (but more moderate than NorESM2-LM) (Fig. 6e). It is noteworthy that the
strong summer variability can also be seen in the eastern coast of the equatorial Atlantic in NorESM2-MM, which is
observed but not simulated in other NorESM runs (Fig. 6a-d). The performance in simulating the seasonal cycle of the
variability is summarized in a Taylor diagram in Fig. 6f. The physical bias correction and updated version of NorESM
improve the SST variability with respect to the reference NorESM1-CTL in terms of seasonality (better correlation). While
NorESM2 is better than NorESM1-AC in terms of correlation, NorESM2-LM has a higher RMSE due to too-strong
amplitude of the summer peak.

To investigate the marine biogeochemical response to the AZM, the Atlantic Niño and Niña events are estimated by
detrending the Atlantic 3 index (det-ATL3) defined as June-July SST anomalies averaged in 20°W-0° and 3°S and 3°N.
From the det-ATL3, the Atlantic Niño and Niña are defined as the det-ATL3 larger and smaller than ± one standard
deviation. Note that 0.75×standard deviation is used as the threshold for observation. Since the monthly primary production
data is only available from 2000 to 2019 and the Atlantic Niño/Niña tends to be weaker during these decades (e.g., Prigent et
al., 2020), the lower threshold yields more events of Atlantic Niño and Niña events. The events in NorESM simulations are
defined by the individual ensemble member's climatology and standard deviation. To emphasize the anomalies due to the
Atlantic Niño, the difference in composite between Atlantic Niño and Niña are shown and the values of composite anomalies
are scaled by ATL3 index in the observation and simulations.

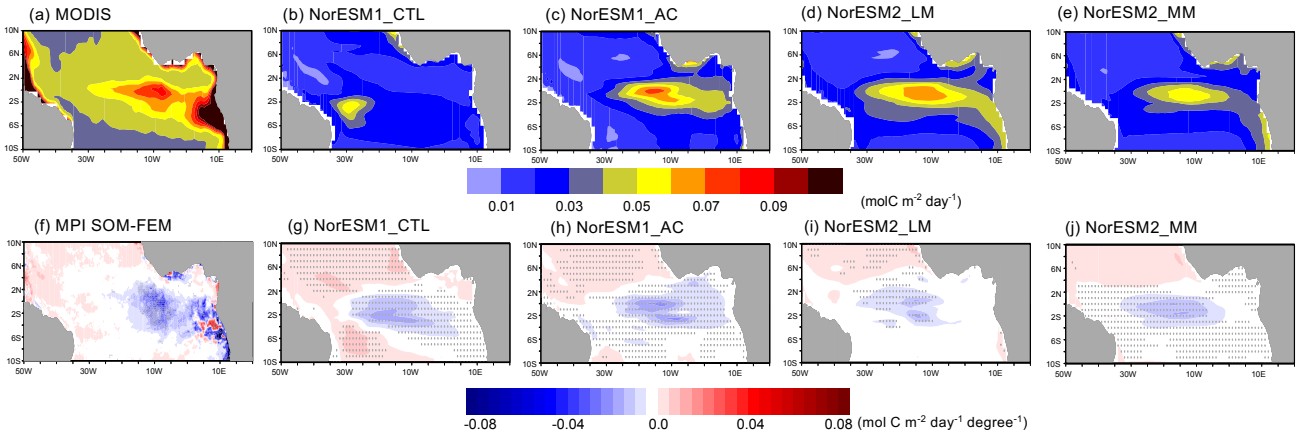

**Figure 7:** June-July-mean primary production for (top) climatology and (bottom) composite anomalies between Atlantic Niño and Niña
for the observations and each NorESM simulation. The composite anomalies are scaled by ATL3-index anomalies between Atlantic Niño
and Niña. Grey dots denote significance level of 90% estimated by Student's *t*-test.




In the observed climatology in June and July, the high productivity extends from the African coast to the equatorial Atlantic (Fig. 7a, see also Fig. 3f). The primary production is suppressed during the Atlantic Niño around 15°W to 10° W at the equator (Fig. 7f) (around the African coast, there are stronger but less significant anomalies). Chenillat et al. (2021) suggested that chlorophyll-*a* variability is driven mainly by the upwelling associated with Atlantic Niño and the corresponding nitrate supply from the ocean subsurface. NorESM1-CTL fails to reproduce the observed climatological Atlantic summer bloom and the maximum of primary production located closely to the northeastern Brazilian coast with a smaller magnitude (Fig. 7b). The strong suppression of the primary production during Atlantic Niño is located erroneously around 20°W, which is relatively westward from the observation (Figs. 7f and g). As shown in Fig. S5, the primary production anomaly during the Atlantic Niño is much worse than those during the Atlantic Niña. With the physical bias correction, the core of the Atlantic summer bloom is located in the central equatorial Atlantic (Fig. 7c) and the reduced primary production anomaly have a peak around 10°W, which is more realistic, in NorESM1-AC (Fig. 7h). Compared to NorESM1-CTL, the climatology and ATL3-scaled response of primary production is larger in NorESM1-AC, which is more in line with the observation (Fig. 7g and h). NorESM2 configurations also simulate the summer bloom at the more realistic location elongating from the eastern to central basin although the magnitude of the bloom is underestimated (Fig. 7d and e). In addition, there is some productivity (much smaller than the observation) along the western African coast (5°S to 10°S) that NorESM1s fail to reproduce. This could be associated with the riverine flux implemented in NorESM2s (Tjiputra et al., 2020). The suppression of primary production associated with the Atlantic Niño is well captured in the central basin (20°-10°W) at the equator, but its amplitude in NorESM2-LM is relatively smaller than NorESM1-AC (Fig. 7i). In NorESM2-MM, the climatological primary production is better reproduced with a larger amplitude than that of NorESM2-LM (Fig. 7d and e). The suppression of primary production is captured in the central basin at the equator during the Atlantic Niños (Fig. 7j).

As Chenillat et al. (2021) showed, the primary production during the summer fluctuates predominantly due to anomalous upwelling, modulating the nutrient supply from the subsurface, associated with the Atlantic Niño and Niña events. In NorESM1-CTL, the supply of nitrate is reduced during the Atlantic Niño consistent with the suppressed primary production and the anomaly minimum is centered around 100 m depth and 20°W (Fig. 8a). These upwelling-induced nitrate anomalies largely drive the simulated primary production anomalies. Compared to NorESM1-CTL, the nitrate anomalies shift shallower and eastward in NorESM1-AC (Fig. 8b). The negative anomalies crop up just below the ocean surface (~40 to 20m) in the central to eastern basin (20°W to 10°E), which is unclearly seen in NorESM1-CTL. This eastward shift and shoaling of nitrate anomalies appear to be important to produce more comparable primary production anomalies with the observations in NorESM1-AC than in NorESM1-CTL (Fig. 7g and h; e.g., the primary production in the model occurs in the euphotic zone fixed to the top 100m depth). Similarly, the shallower nitrate anomalies in NorESM2s are located in the central to eastern basin in Fig. 8c and d. Outcropping of the nitrate anomalies to the near-surface is also detected and



consequently, the primary production anomalies are comparable with the observations, especially in terms of location (Fig. 7i and j).

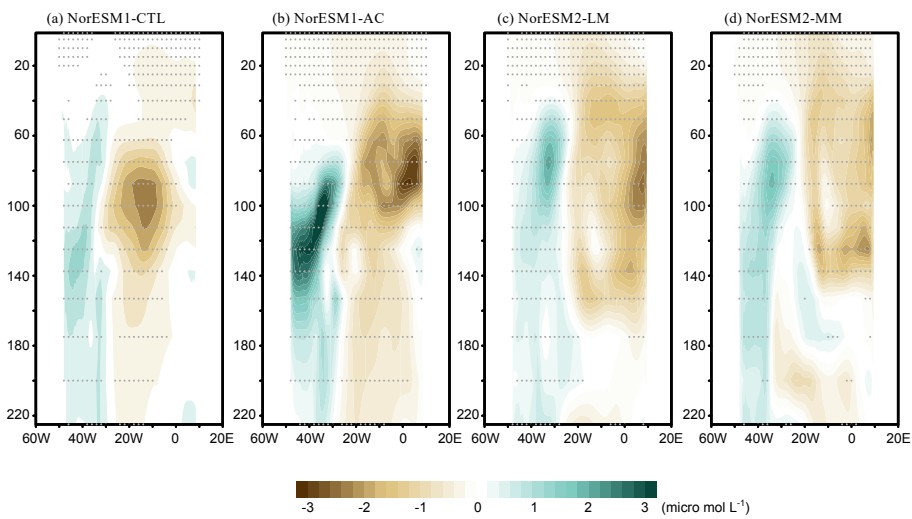

**Figure 8:** Depth-longitudinal sector (averaged between 3°S and 3°N) of June-July-mean composite anomalies of nitrate concetration between Atlantic Niño and Atlantic Niña in each NorESM simulation. Gray dots denote a significance level of 90% by Student's *t*-test.


The observation shows that the climatological outgassing (ocean-to-atmosphere) $CO_2$ maximum is located in the western basin of the equatorial Atlantic and another moderate peak is detected in the central basin (Fig. 9a). As Koseki et al. (2023) showed, the $CO_2$ flux responds to the Atlantic Niños with a dipole structure in the equatorial Atlantic (Fig. 9f): The $CO_2$ outgassing is reduced during the Atlantic Niños around the northeastern Brazil coast (50°W-30°W), away from the core

of SST anomalies (Fig. 6a and (Koseki, 2023). Contrastingly, the $CO_2$ outgassing is enhanced in the central to eastern basin during the Atlantic Niños. According to Koseki et al. (2023), this dipole structure of anomalies is induced mainly by freshwater (western basin) and SST anomalies (central to eastern basin), which change the surface partial pressure of $CO_2$. The spatial $CO_2$ flux pattern in NorESM1-CTL is largely biased, as shown in Fig. 9b. The climatological flux has its outgassing peak in the central basin more southward and there is a weak $CO_2$ uptake around the northeastern coast of Brazil

(Fig. 9b). An ingassing bias is simulated along the African coast between 10°S and 6°S. NorESM1-CTL also fails to reproduce the spatial pattern of flux anomalies associated with the Atlantic Niños (Fig. 9g). The observed dipole structure of $CO_2$ flux anomalies during the Atlantic Niño is incorrectly simulated off the equator between 35°W and 0° at 6°S (Fig. 9f).

The climatological physical bias correction approach implemented in NorESM1-AC is somewhat successful in improving the climatological summer sea-air $CO_2$ flux in Fig. 9c. Although it is overestimated and the maximum of

outgassing shifts southward compared to the observations, the strong upward $CO_2$ flux occurs more realistically in the western basin (Fig. 9c). The uptake bias remains along the west African coast indicating that the $CO_2$ flux variability here is not predominantly driven by SST, but rather by the bias in the biogeochemical properties or by the lack of riverine flux. The





Atlantic-Niño-induced $CO_2$ flux anomalies are generated more realistically along the equator having dipole structures and comparable amplitudes with the observations while their locations are still slightly southward (Fig. 9h). The two versions of

NorESM2 are also successful in simulating the climatological summer $CO_2$ flux in the tropical Atlantic (Fig. 9d and e): the maximum of outgassing $CO_2$ flux is located between 6°S and 0°, which is almost identical with the observations (Fig. 9a) and its amplitude is also more realistic (~1.5 mol C $m^{-2}$ $yr^{-1}$) than NorESM1-AC (Fig. 9c). Additionally, the NorESM2 configurations can alleviate ingassing bias along the African coast as well. The dipole pattern of $CO_2$ flux anomalies is also broadly represented along the equator in NorESM2s (Fig. 9i and j).

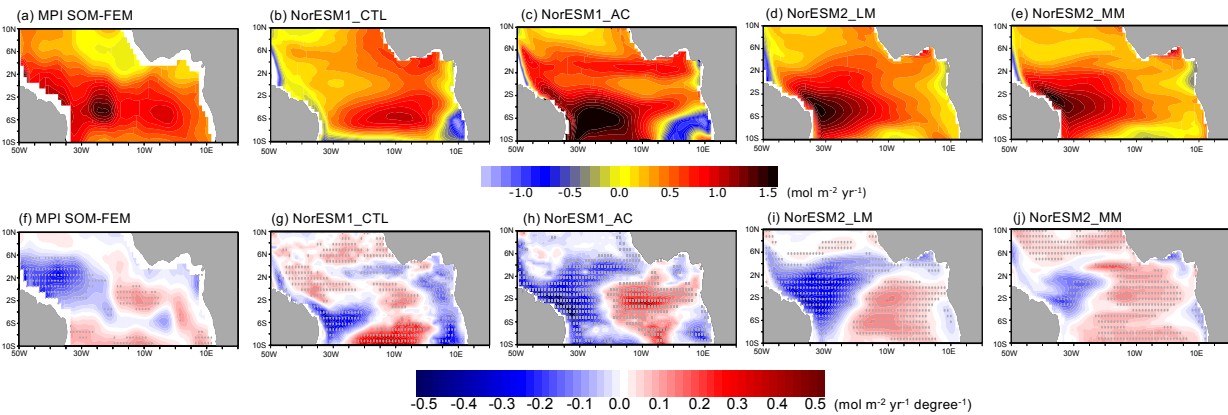

**Figure 9:** June-July-mean surface $CO_2$ flux for (top) climatology, and (bottom) composite anomalies between Atlantic Niño and Atlantic Niña for the observations and each NorESM simulation. Outgassing is shown by positive value. The composite anomalies are scaled by ATL3-index anomalies between Atlantic Niño and Niña. Grey dots denote a significance level of 90% estimated by Student's *t*-test.

The surface ocean $pCO_2$ is one of the main driver of the sea-air $CO_2$ flux (e.g., Sarmiento, 2006). In NorESM1-CTL, the SSS negative anomaly is found in the central to eastern basin during Atlantic Niño covering the ACT whereas the positive anomaly occurs in the north tropical Atlantic (Fig. 10a). This SSS anomaly pattern reflects the displacement of the ITCZ associated with the warm event at the equator. The $CO_2$ flux anomaly pattern appears to be roughly consistent with these SSS anomalies: in the western basin, the less (more) $CO_2$ outgassing corresponds to the negative (positive) SSS at 8°S-

6°S (2°N-4°N). A part of the negative SSS anomalies covering the ACT co-locates with the less $CO_2$ outflux (Fig. 9g).
In NorESM1-AC, the negative SSS anomaly is found mainly in the western basin along the northeastern Brazilian coast and the positive SSS anomaly occurs northward of the negative SSS anomaly (Fig. 10b). As in NorESM1-CTL, this SSS anomaly pattern is associated with the ITCZ southward displacement, but the SSS anomalies are more dominant in the western basin in NorESM1-AC resulting in the less outgassing anomalies of $CO_2$ flux in the western basin, which is more

realistic (Figs. 9f and h). This difference in the ITCZ displacement and corresponding SST anomalies derive from the realistic development of the ACT during summer between NorESM1-CTL and NorESM1-AC (Fig. S2). In NorESM1-CTL, the ACT hardly develops and the climatological ITCZ is anchored more southward than the observation (e.g., Koseki et al.,



2018) and consequently, the ITCZ is perturbed by the Atlantic Niño around the equator. In the two NorESM2 versions, the SSS negative anomalies are also dominated in the western basin (Figs. 10c and d) and the $CO_2$ flux is correspondingly

reduced in the western basin at the equator (Figs. 9i and j). Both NorESM2 simulations also reproduce the summer ACT development more realistically than NorESM1-CTL (Figs. S2c and d) and the freshwater anomalous inputs associated with the ITCZ displacement can be well captured resulting in the reduction of the $CO_2$ flux in the western basin.

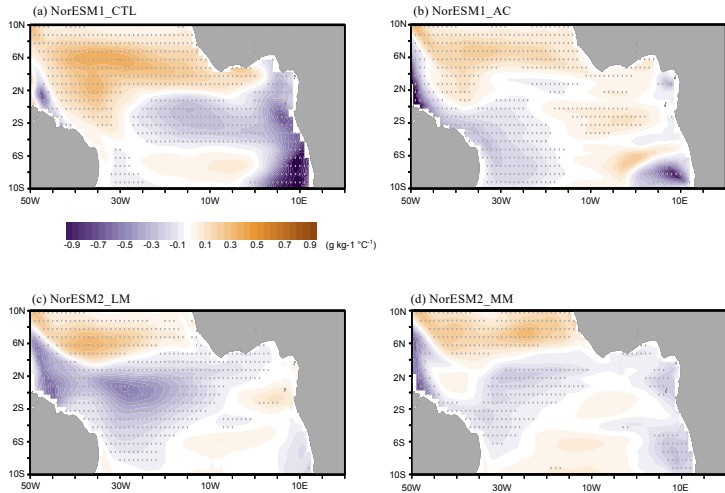

**Figure 10:** June-July-mean composite anomalies of sea surface salinity for each NorESM simulation. The composite anomalies are scaled
by ATL3-index anomalies between Atlantic Niño and Niña. Gray dots denote a significance level of 90% by Student's *t*-test.

## 4 Summary and Discussion

This study evaluated implications of physical bias on the simulated marine biogeochemical processes in the tropical Atlantic Ocean for 4 different realizations of the NorESM. A physical bias correction and better dynamical representations in new generation of NorESM improve the tropical Atlantic physical and biogeochemical biases during boreal summer, which

are common in other ESMs (e.g., Voldoire et al., 2019). The seasonal development of the Atlantic Cold Tongue (ACT) is simulated more realistically during the boreal summer in NorESM1-AC and the NorESM2s than in the benchmark simulation of NorESM1-CTL. Associated with the better ACT development, the observed zonally-tilting thermocline is also well reproduced. NorESM2s can reproduce the shoaling in the eastern basin without any bias correction. This improvement of the thermocline gradient leads to a better representation of the observed nutrient supply from the subsurface in the eastern

basin. Consequently, NorESM-AC and NorESM2s can simulate the observed timing (July to September) and location (centered at 10°W along the equator) of the Atlantic summer bloom of primary production. While NorESM2s include updates and tunings of physical and biogeochemical parameters relative to NorESM1s (e.g., (Ilicak et al., 2008; Tjiputra et al., 2020; Toniazzo et al., 2020)), NorESM1-AC only implements physical bias correction of surface wind and SST, which also resulted in remarkable improvements in its mean state and variability of biogeochemical processes. Our results



emphasize that atmospheric and ocean dynamics/physics are crucially important to simulate regional marine biogeochemical processes and their interaction in the tropical Atlantic (e.g., Berline et al., 2007; Fransner et al., 2020).

The benefit of physical bias correction can be especially seen along the Benguela upwelling region, where the highest biological production is observed in the tropical-subtropical Atlantic (e.g., (Shannon et al., 2004). With the physical bias correction, the high production area is confined along the Angola-Benguela coast, alleviating the initially
underestimated biological production (Fig. S4). This is attributed to the better upwelling and nutrients supply (Fig. S3) associated with the corrected coastal low-level jet and wind stress curl that are essential drivers of coastal upwelling (e.g., Koseki et al., 2018; Lima et al., 2019). Contrastingly, NorESM2s tend to degrade the coastal production in the southeast Atlantic. This might be due to the tunings of biological parameters to reduce the largely-overestimated production in other ocean areas (Tjiputra et al., 2020). However, due to the newly-implemented riverine flux (Gao et al., 2023), the primary
production is to some extent enhanced around the Congo river mouth (around 5°S) as compared to the NorESM1 (Fig. S4), which does not include riverine flux. Between NorESM2-LM and NorESM2-MM, the SST bias and nutrients upwelling biases are alleviated in NorESM2-MM where the atmospheric component resolution is finer than that in NorESM2-LM. The atmospheric refinement is beneficial to improve the model performance in reproducing the tropical Atlantic climate (Harlass et al., 2018).

With better representation of the physical processes, the interannual variability of biogeochemical processes is also improved. As Chenillat et al. (2021) showed, the Atlantic Niño is one of the essential drivers for variability in the primary production in the equatorial Atlantic. NorESM-AC and NorESM2s can reproduce the reduction of the summer bloom in the central basin while NorESM-CTL simulates the summer bloom anomaly in the wrong location. Because the primary production anomaly is mainly induced by the upwelling modulation associated with the Atlantic Niño (e.g., Chenillat et al.,
2021), a more realistic thermocline structure in NorESM-AC and NorESM2s is able to capture the observed summer bloom variations. The sea-air $CO_2$ flux anomalies associated with the Atlantic Niño are also more realistically reproduced in NorESM2-AC and NorESM2s than NorESM1-CTL. The $CO_2$ flux anomalies in the western basin is mainly driven by the SSS anomalies associated with the ITCZ displacement (Koseki et al., 2023) and this study suggests that the realistic representation of the ACT and ITCZ are responsible for simulating the observed $CO_2$ flux anomalies due to the Atlantic
Niño. We also note that in addition to proper physical representation, accurate representation of subsurface biogeochemical state is also crucial in reproducing the observed variability in an upwelling system such as the tropical Atlantic (e.g., Ayar et al., 2022; Koseki et al., 2023).

The physical bias is one of the main reasons why the climate prediction and projection are uncertain (e.g., Bethke et al., 2021; Counillon et al., 2021; Crespo et al., 2022). As we showed in this study, the physical bias reduction allows us to
reproduce more realistic marine biogeochemical processes by improving interaction between physics and biogeochemistry. Therefore, future improvements in biogeochemical processes and parameterization (Singh et al., 2022; Tjiputra et al., 2007) should also take into consideration biases in physical processes to avoid overfitting or correctly simulating biogeochemical processes but for wrong reason. Our study also highlights the importance of evaluating the Earth system models'



performance at regional scale and at timescale where natural climatic variability dominates over external forcing. Improvements at these spatial and temporal scales are particularly valuable due to the more direct and significant impacts on the society. Future model evaluation should go beyond capturing the large scale, mean state features and focus more on regional dynamics across seasonal-to-decadal time scales.

**Acknowledgement**

This study is supported by the H2020 TRIATLAS project (grant# 817578). The observational data of SST and $CO_2$ flux are available to download at https://www.ncei.noaa.gov/products/optimum-interpolation-sst and https://www.ncei.noaa.gov/access/ocean-carbon-data-system/oceans/MPI-ULB-SOM_FFN_clim.html, respectively. The computational resources of NorESM simulation and data archive are supported by UNINETT Sigma2 AS (NN9039K and NS9560K).

*Authors Contribution*

SK and LRC have conducted NorESM1 simulations and conducted the analysis of the experiments. All the authors have discussions on results and their interpretations. All the authors have contributed to build a manuscript and improve it to the final form of the manuscript.

*Data Availability*

All data is available on request.

*Code Availability*

All codes are available on request.

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
