# Peer review of "Assessing the tropical Atlantic biogeochemical processes in the Norwegian Earth System Models"

_EGUsphere, 2023_

## Author Comment (AC2)

**Reply to Reviewer #2**

This study analyses the skills of the ESM NorESM at reproducing the physical and biogeochemical characteristics of the tropical Atlantic Ocean. A set of 4 model configurations is compared: the NORESM1 configuration, the NORESM1 configuration with flux correction based on observations, and two configurations based on NORESM2 (coarse and medium spatial resolution of the atmospheric component). The standard NORESM1 setup exhibits strong biases both on ocean dynamics and biogeochemistry which are improved with flux correction or a higher resolution of the atmospheric configuration. The low resolution version of NORESM1 also shows some improvements which suggests that the new parameterizations and calibration in the version 2 also contribute to the improved skills. Another important conclusion of the study is that biases in the simulated ocean dynamics have a strong imprint on the simulated ocean biogeochemistry (nutrients, NPP and pCO2). And this concerns the mean state, the seasonality and interannual variability.

**REPLY**: We grately appreciate the reviewer for reviewing our manuscript very carefull and providing constructive comments. Here, we reply to the comments point-by-point and will upload the revised manuscript following these comments. Please note that any revisions in the manuscript will be given in **blue-color font** for easy-tracking.

**I have several general concerns:**

**1) The description of the different model configurations and the main differences between them is rather short and is thus very difficult to follow for someone who is not an expert of NorESM. I think that some additional information is necessary such as more details on the flux correction technique, on the main differences between the different model components that are relevant for the study.**

**REPLY**: Thank you very much for this comment. We added some texts on anomaly coupling method and made a table of four configiuraitons of NorESM we used in this study as shown below. This table will be added as Table R1 and give some description in the **Section 2.2: Model configurations** . Please see lines at 120-122 and 136-137.

| | Atmosphere | Ocean | Bias Correction | New Paramerization / Updates | Ensemble Number | Historical Period |
|---|---|---|---|---|---|---|
| NorESM1-CTL | CAM4 (143x96) | MICOM (319x384) | No | No | 5 | 1990-2019 |
| NorESM1-AC | CAM4 (143x96) | MICOM (319x384) | Anomaly Coupling (Toniazzo and Koseki, 2018; Counillon et al., 2021) | No | | 1990-2019 |

| NorESM2-LM | CAM5 (143x96) | BLOM (319x384) | No | Ocean mixig layer (Ilicak et al., 2009)  Ocean eddy diffusion (Eden et al.,2009)  Atmospheric angular momemtum (Toniazzo et al.,2020)  More details, Seland et al. (2020) | 3 | 1990-2014 |
|---|---|---|---|---|---|---|
| NorESM2-MM | CAM5 (287x192) | BLOM (319x384) | No | Ocean mixig layer (Ilicak et al., 2009)  Ocean eddy diffusion (Eden et al.,2009)  Atmospheric angular momemtum (Toniazzo et al.,2020)  More details, Seland et al. (2020) | 3 | 1990-2014 |

**Table R1.** List of the atmospheric and oceanic components and their spatial horizontal resolution for the four different NorESM configurations. Information on bias correction, parameterization/updates of each component, ensemble number, and the historical period analyzed in this study are also provided.

**2) The different configurations are quite well designed to illustrate the improvements to be expected, at least from a better representation of the atmospheric state (flux correction, higher spatial resolution). However, I find that the attribution and the mechanistic understanding are rather too vague. For sure, some changes in the model components explain part of the improvement since NorESM2-LM performs better than NorESM1-CTL, but there is no discussion on these changes and what role they play in the improvement. For instance, what is explained by changes in the atmopheric, oceanic and biogeochemical components respectively? This would probably require additional experiments such as a NorESM2-AC. We also see that the upper thermocline is much more stratified in NorESM1 than in NorESM2: why and what are the consequences? Winds and evaporation minus precipitation are the main players but we have no idea of the biases they exhibit in the non corrected model configurations. A consequence of this general concern is the discussion that is really vague and not very informative, to my opinion.**

**REPLY**: Thank you very much for raising this important point. Regaridng a new experiment like **NorESM2-AC**, we agree that it is worth of performing bias-corrected simulations with NorESM2. On the other hand, as our results show (Figs. 1 and 2), NorESM2 simulations are successful to reduce the tropical Atlantic biases, which is more or less comprable with NorESM1-AC and our study is a first one to investigate and assess NorESM2 in the tropical Atlantic biogeochemical processes. Therefore, we intended to focus on NorESM2 without any corrections. In order to bias correct

NorESM2, we would need more compitational time and other resources, due to the more complex components, to implement the method of anomaly coupling (Toniazzo and Koseki, 2018): modifications of model's source codes, spin-up with anomaly coupling, etc. Therefore, we consider that bias -correction of NorESM2 will be more suitable for future works. We will add this discussion in the **Summary and Discussion**.

However, we agree with the reviewer that more clarification on why NorESM2 simulations can reduce the bias of the tropical Atlantic are warranted. As we described in the Section2, there are simultaneous updates of physical and biogeochemical parameterizations included in NorESM2 from NorESM1, in addition to updated atmospheric and land components. Therefore, it is not feasible to isolate which new paramerization or process improvement is responsible for the improvements in the ocean biogeochemistry. Therefore, as a first order, we examinaed the vertical strucutre of ocean temperature as in Fig.2. The vertical strucuture is fundamental to investigate the model bias in the tropical Atlantic and NorESM2 expeirments have better zonal gradient of the thermocline. This is an indication that ocean physical processes such as upwelling and Kelvin wave propagation (responsible for the thermocline gradient) are improved in the NorESM2. To make this implication more robust, we compare the seasonal cycle of sea surface height (SSH) among the observation and NorESM simulations as shown below in Fig.R1.

[Figure]

**Figure R1.** Hovmöller plot of sea surface heoght anomaly from annual mean (averaged between 3S-3N). The contour is for AVISO obsrvation (countoru interval is 0.01m, 1993-2001) and color shading are from different NorESM simulations.

In NorESM1-CTL, the seasonal cycle of high and low SSH in the eastern basin (20W-10E) is delayed by 1-2 months as compared to the observation. This is a common bias of ESMs generating a warm bias and a misrepresentation of the Atlantic Cold Tongue.

Especially, the low SSH is a result of eastward propagation of upwelling Kelvin wave from spring to summer. The westerly trade wind is a key driver of the upwelling Kelvin wave.

In NorESM1-AC, the climatological bias correction leads to considerable improvements of the SSH seasonal cycle. Applying this methodology of bias correction, the ocean component is forced by the right surface wind and this indicates that the upwelling Kelvin wave is also realistically generated. This appears to be the primary reason for the improved marine biogeochemical processes in NorESM1-AC.

In NorESM2, the seasonal cycle of SSH remains biased (in particular, NorESM2-LM), however, the shoaling of SSH is more realistic than NorESM1-CTL, for example, the shallowest longitude is 10E (as the observation) in NorESM2 and 10W-0 in NorESM1-CTL. This indicates that the upwelling Kelvin wave propagation is represented better in NorESM2 than in NorESM1-CTL. Actuallty, the dynamics among surface wind, Kelvin wave, SSH, and SST is maintained by the Bjerknes Feedback and it is hard to quantify which components play a more important role in alleviating the bias. However, we can indicate that NorESM2 reproduce more realistic air-sea interaction than NorESM1-CTL as shown in Figs. 2 and R1 and consequently, the marine biogeochemical processes are also improved as shown in Fig.5. We have added this statement and Fig.R1 as new Fig.S3. Please see lines at 198-204.

Regarding the stratification, one of possible causes might be different ocean mixing layer parametertization between NorESM1 and NorESM2. Other possobility is ocean circulation at deeper layer. As Figs. 2 and 5 show, NorESM1 has much cooler subsurface ocean and more nutrients. This indicates that upwelling of deeper ocean watermass is stronger in NorESM1. We also note that there might be other drivers for this difference, for example, the stronger AMOC in NorESM1. We already discussed this point at lines 278-280 in the original manuscript.

**3) Marine biogeochemistry is evaluated by inspecting nutrients, PP and pCO2. pCO2 is very sensitive to the dynamics (as mentioned and shown in the study) and it is thus not very surprising that any improvement in the representation of ocean dynamics has a strong impact on it. It is not a very good tracer of the ecosystem component of the biogeochemical model. PP is not observed but reconstructed from some algorithms both for chlorophyll and PP itself which are known to have significant issues (different algorithms can give very different results). I would have liked to see a comparison to chlorophyll satellite data which are much more direct and with less uncertainties.**

**REPLY**: Thank you very much for raising this important point out. Yes, Chlorophyll should be an interesting variable to investigate. However, the biogeochemical component of our models is relatively simple and then, there is no output of chlorophyll. Primary production we analyzed here is a diagnostic variable of NorESM based on concentration of Phythoplankton (there is no trophic level), which is a

prognostic variable. In the revised manuscript, we will add the analysis of Chrolophyll calibrated by NorESM Phytoplankton and Level-4 observed chrolophyll data.

**In summary, I think that this study needs some major revisions addressing my general concerns before it can deserve publication. A crucial point is a more thorough investigation of the features that explain the improvements obtained in the different model configurations. Finally, the model performs quite bad in terms of PP and pCO2, even in the best configurations depite what the authors state sometimes in the study. However, this is not a concern for me because ESM but also quite coarse ocean-only models tend to behave quite badly in this basin. However, I would be curious to see Chlorophyll.**

**REPLY**: Thank you very much for your constructive comment. As we answered to the previous comment, our model does not have Chlorophll as outputs and then, primary production might be a good comparison with the observation as a first choice.

**Minor comments:**

**on the manuscript as a whole: Obviously I'm not a native English speaker, but I think the English can be improved. In addition, there are typos and formatting problems with references throughout the manuscript that should be corrected.**

**REPLY**: Thank you very much for the comment. We have read the manuscript again more carefully and corrected the grammer and typos.

**Section 2.3: You don't explain what MPI SOM-FEM is.**

**REPLY**: Thank you very much for the comment. We have added it.

**Line 176: NorESM2 has a warmer subsurface and a less stratified upper thermocline (seen also on the nutrient vertical distribution), why? It relates to my general concern 2.**

**REPLY**: As we answered to the previous comment, this might be because of the difference in the AMOC between NorESM1 and NorESM2. This statement is already given in the original manuscript. Please see lines at 278-280.

**Section 3.2 and figure 3: The ACT is clearly improved, especially in NorESM2-MM but also in NorESM2-LM and is better (at least from what I can see) than NorESM1-CTL. Thus, part of the improvement is not related to the increased**

atmospheric resolution but to changes in the ingredients of the physical components. Any clue on what they are. Furthermore from January to June, NorESM2 is not that good and worse than NorESM1-CTL. It is significantly warmer and with two maxiam close to the African coast and 30-35W. It should be mentionend and ideally commented.

**REPLY**: Thank you very much for this comment. As we replied to the previous comment, we have added an analysis on sea surface height (SSH) seasonal cycle in Fig. R1 (new Fig. S3 in the revised manuscript). Actually, NorESM2-LM/MM have better shoaloing in the eastern basin of the equatorial Atlantic during summer. This indicate that NorESM2 has better ocean physics than NorESM1-CTL. In addition, NorESM2-MM has better seasonal cycle in SSH in summer. This statement is given in the revised manuscript. Please see lines at and new Fig.S3.

Regarding the warmer SST in the western basin, we have added some texts on it. Please see lines at .198-199.

**Lines 262-264: what are these improvements? Very vague.**

**REPLY**: Here, we mention the difference in vertical strucutre of ocean temeprature and nitrate concentration and possible causality of the difference.

**Section 3.3: Why using different types of analysis for SST and PP? Is there any reason behind the differential treatment?**

**REPLY**: It has been shown ealier (Chenilatt et al. (2021) and Koseki et al. (2023)) that primary production (Chlorophll-a) and sea-air $CO_2$ flux respond sensitively to the inter-annual variability of SST, in particular, Atlantic Niño during summer. Therefore, following their findings, we think it's crucial to first explore how well the NorESM simulations reproduce the Atlantic Niño in terms of its intensity, peak time in summer, and location. This can be assessed in Fig. 6. Following this evaluation, it is important to investigate how the marine biogeochemical processes are influenced by the Atlantic Niño and Niña. Primary production and sea-air $CO_2$ flux are also determined by other physical and biogeochemical processes that are fluctuated directly and indirectly by the SST anomalies. Therefore, in Figs. 8 and 9, we explore the performance of key marine biogeochemical process fluctuations in response to the Atlantic Niño/Niña. In the case of primary production, upwelling of nitrate is the main drivers, consistent with Cheniallt et al. (2021), while parts of sea-air $CO_2$ flux response is determined by changes in sea surface salinity (Koseki et al., 2023).

**Why not a taylor plot for PP (or even better Chlorophyll) similar to what is done with SST.**

**REPLY**: Even though the seasonal cycle of primary production improved following the better SST, it is not necessary that the Taylor plot between two variables becomes quite similar because SST is only one of functions/indicators to determine the primary production.

**Lines 380-382: the ingassing bias between 8S and 10S along Africa is quite strong in NorESM1. What is the cause of that sink. PP does not seem to be very very high at the specific location according to Figure S4. In lines 386-387, it is stated that it might be biogeochemical issues or riverine input. This is really vague and does not say anything.**

**REPLY**: Because NorESM1-AC also reduces the SST bias along the African coast, this ingassing bias in both NorESM1-CTL and NorESM1-AC might stem from marine biogeochemical process associated with the riverine flux, which is one of main differences in biogeochemical processes between NorESM1 and NorESM2. Other possibility could be freshwater input from the Congo River. As Awo et al. (2022) showed, the SSS along the coast is influenced by Congo River plume using a high-resolution regional ocean model. There might be too much of freshwater input from the Congo River. On the other hand, our model has a coarse resolution and it is not adequate to investigate the coastal region. We have added this statement. Please see lines at 413-417.

---

## Author Response (AR1)

**Reply to Reviewer #1**

**REPLY:** We grately appreciate the reviewer for reviewing our manuscript very carefull and providing constructive comments. Here, we would like to reply to the comments point-by-point and will upload the revised manuscript following these comments. All replies are are given in **blue-color font** for easy-tracking and line numbers are for the revised manuscript.

**General comments**

This study evaluate the implications of physical biases on the simulated marine biogeochemical processes in the tropical Atlantic Ocean for 4 different version of a ESM. The models used are different versions of NorESM, an earth system model with different components, with an increasing degree of complexity and resolution. The different results are compared to a base solution, NorESM1, taken as the benchmark.

The main improvement was to decrease the bias of annual mean of SST, giving rise to a realistic development of the Atlantic Cold Tongue (in geographical location and timing), and hence the marine primary production in the Equatorial Atlantic ocean. This shows the clear link between the physical cycles and the biological ones. Consequence of the improvements in the physical representations of the system, is also the improvement of the carbon cycle representations, discussed in the manuscript mainly in terms of air-sea C02 fluxes.

The development of the manuscript start by a broad review of the oceanography of the tropical Atlantic ocean, including it's links with coastal phenomena (river inputs), the circulation in neighboring tropical systems, and characteristics phenomena of variability in the region (Atlantic Niño's), and the consequences in terms of anthropogenic and global change effects. The role of ESM is also introduced as key tools, as well as the importance of the physical phenomena on the biogeochemical cycles. Their biases in the physical components clearly decreases its performance downstream regarding the biogeochemical cycles (primary and secondary, oxygen, carbon).

Within this problematic issues, the present manuscript introduce the physical, biological and chemical components of the several versions of the NorESM configurations, and analyze the improvements with relation to the base model, concerning the mean annual, the seasonal and inter-annual time scales.

The NorESM model contributes to CMIP (5 and 6), which provide a degree of general quality and confidence on the results. However, for someone not necessarily familiar with global scale model analysis and its limitations, the large bias reported, even in the most recent (with better performance) versions,

give reasons for some degree of concern regarding the confidence for simulations for the recent past / present and mainly the future scenarios.

The structure of the results starts from the comparison with climatological standard data, and the reasons to induce so large bias, primarily associated to wind stresses and air-sea fluxes in the atmospheric components. The improvements of the different versions justify its application, in terms of horizontal and vertical distributions (Figures 1 and 2).

The seasonality is analyzed along the equator in terms of SST, primary production and PCO2 when compared to the climatological values, (Fig 3) and a thorough analysis (although a bit 'too verbose') of the differences and the improvements was done in the manuscript.

The next step was to analyze the interannual variability, dominated by Atlantic Niño/Niña phenomena. One wonders if the models are able (or not) to reproduce the actual Niño/a's years in the recent pass (I think that the response is probably not), as the forcing used in the most advanced models should include the atmospheric mechanisms (wind stress anomalies) to start Niño/a(s). I think that some comment should be done around this issue.

**REPLY**: We grately appreciate Reviewer#1 for reviewing our manuscript very carefully and providing constructive comments. Since Earth system models simulate their own internal climate variability, they are (and should) not reproduce the actual Nino/a's years. Nevertheless, in terms of reproducing the general characteristic of the observed Atlantic Niño/Niña, the reviewer is correct that even state-of-the-art prediction models that are initialiuzed with observation still have numerous challenges to overcome. However, as Fig.R1 shows (obtained from Counillon et al., 2021) that physical bias correction, which was also employed in our study (ACPL) has a potential to alleviate the low prediction skill of Atlantic Niño index in the equatorial Atlantic (please note that our models are identical with Counillon et al., 2021). In this prediction system, sea surface temperature anomaly is initialized.

[Figure]

**FigR1**. Prediction skill of ATL3 index (20W-0 and 3S-3N) performed by NorESM1-CTL (blue) and NorESM1-AC (red). Obtained from Counillon et al. (2021). Values close to one indicate high predictive skills. X-axis denotes leading time (month).

While the paper showed the improvement in predictive skill, May-initialized prediction has relatively lower improvement, indicating that the prediction of Atlantic Niño/Niña and the corresponding marine biogeochemical processes is far from satisfactory. We added this discussion in the **Summary and Discussion**. Please see lines at 496-499.

**The analysis centered the attention around the STD of several fields, (Fig 6 ), composite anomaly differences in the horizontal (Fig 7) and in vertical sections (Fig 8 and 9) for different variables. It seems to me a too technical and specialized explanation section for modelers, while I would expect some comments within the discussion section about this important issue.**

**REPLY**: The motivation for our analysis is (1) because the state-of-the-art ESMs in CMIP6 still have large uncertainties not only in the simulated mean climate state, but also in inter-annual variability in the tropical Atlantic, it is important to investigate the mechanistic drivers of these uncertainties (here, we tested different configurations of a single model as our methodological approach) and (2) because recent studies show impacts of the Atlantic Niño on marine biogeochemical processes like Chrolophyll (Cheniatt et al, 2021) and sea-air $CO_2$ flux (Koseki et al., 2023). It is therefore necessary to assess how the limitations of the machanisms identified in (1) affect the region's marine biogeochemistry. Because it seems that we did not clearly mention part of these motivations in the manuscript, we added the motivation in the beginning of the Section 3.3. Please see lines at 315-317 and 318-320.

**Otherwise the manuscript are well organized and well written, and deserves to be published in my opinion.**

**REPLY**: Again, thank you very much for the reviewer's constructive comments and positive assessments.

**Specific comments**

**The description of the different versions of NorESM model is rather difficult to follow for someone that does not know the NorESM\* system, and a table containing the four versions and main features would help to the reader better identify the common points and differences between models.**

**REPLY**: Thank you very much for this suggestion. We have added a table of four configiurations of NorESM we used in this study as shown below. This table is added as Table 1 and description of it has been added in the **Section 2.2: Model configurations**. Please see lines at 134-135.

| | Atmosphere | Ocean | Bias Correction | New Paramerization / Updates (Physics) | New Paramerization / Updates (Biogeochemistry) | Ensemble Number | Historical Period |
|---|---|---|---|---|---|---|---|
| NorESM1-CTL | CAM4 (143x96) | MICOM (319x384) | No | No | No (HAMOCC, Tjiputra et al., 2013) | 5 | 1990-2019 |
| NorESM1-AC | CAM4 (143x96) | MICOM (319x384) | Anomaly Coupling (Toniazzo and Koseki, 2018; Counillon et al., 2021) | No | No (HAMOCC, Tjiputra et al., 2013) | 5 | 1990-2019 |
| NorESM2-LM | CAM5 (143x96) | BLOM (319x384) | No | • Ocean mixing layer
• Ocean eddy diffusion
• Atmospheric angular momentum

More details in Seland et al. (2020) | • Riverine flux
• Air-sea gas exchange
• Ecosystem parameters adjustments

More details in Tjiputra et al. (2020) | 3 | 1990-2014 |
| NorESM2-MM | CAM5 (287x192) | BLOM (319x384) | No | Same as NorESM2-LM | Same as NorESM2-LM | 3 | 1990-2014 |

**Table 1**. List of four different configurations of NorESM simulation, including the atmospheric and oceanic components and horizontal grid strucutures.

**From my point of view the way how the Figure 4 , containing Taylor diagrams of the SST, PP and CO2 fluxes was done, should be better explained.**

**REPLY**: Thank you very much for this comments. We improve the description and explanation of Fig. 4. Please see lines at 258-260 and 263-267.

**Reply to Reviewer #2**

**This study analyses the skills of the ESM NorESM at reproducing the physical and biogeochemical characteristics of the tropical Atlantic Ocean. A set of 4 model configurations is compared: the NORESM1 configuration, the NORESM1 configuration with flux correction based on observations, and two configurations based on NORESM2 (coarse and medium spatial resolution of the atmospheric component). The standard NORESM1 setup exhibits strong biases both on ocean dynamics and biogeochemistry which are improved with flux correction or a higher resolution of the atmospheric configuration. The low resolution version of NORESM1 also shows some improvements which suggests that the new parameterizations and calibration in the version 2 also contribute to the improved skills. Another important conclusion of the study is that biases in the simulated ocean dynamics have a strong imprint on the simulated ocean biogeochemistry (nutrients, NPP and pCO2). And this concerns the mean state, the seasonality and interannual variability.**

**REPLY**: We grately appreciate Reviewer#2 for reviewing our manuscript very carefully and providing constructive comments. Here, we reply to the comments point-by-point and will upload the revised manuscript following these comments. All replies are given in **blue-color font** for easy-tracking.

**I have several general concerns:**

**1) The description of the different model configurations and the main differences between them is rather short and is thus very difficult to follow for someone who is not an expert of NorESM. I think that some additional information is necessary such as more details on the flux correction technique, on the main differences between the different model components that are relevant for the study.**

**REPLY**: Thank you very much for this comment. We added some texts on anomaly coupling method and made a table of four configiuraitons of NorESM we used in this study as shown below. This table will be added as Table R1 and described in the **Section 2.2: Model configurations** . Please see lines at 119-121 and 134-135.

| | Atmosphere | Ocean | Bias Correction | New Paramerization / Updates | Ensemble Number | Historical Period |
|---|---|---|---|---|---|---|
| NorESM1-CTL | CAM4 (143x96) | MICOM (319x384) | No | No | 5 | 1990-2019 |
| NorESM1-AC | CAM4 (143x96) | MICOM (319x384) | Anomaly Coupling (Toniazzo and Koseki, 2018; Counillon et al., 2021) | No | 5 | 1990-2019 |

| NorESM2-LM | CAM5 (143x96) | BLOM (319x384) | No | Ocean mixing layer (Ilicak et al., 2009)

Ocean eddy diffusion (Eden et al.,2009)

Atmospheric angular momentum (Toniazzo et al.,2020)

More details in Seland et al. (2020) | 3 | 1990-2014 |
|---|---|---|---|---|---|---|
| NorESM2-MM | CAM5 (287x192) | BLOM (319x384) | No | Same as in NorESM2-LM | 3 | 1990-2014 |

**Table R1**. List of four different configurations of NorESM simulation, including the atmospheric and oceanic components and horizontal grid strucutures.

**2) The different configurations are quite well designed to illustrate the improvements to be expected, at least from a better representation of the atmospheric state (flux correction, higher spatial resolution). However, I find that the attribution and the mechanistic understanding are rather too vague. For sure, some changes in the model components explain part of the improvement since NorESM2-LM performs better than NorESM1-CTL, but there is no discussion on these changes and what role they play in the improvement. For instance, what is explained by changes in the atmoperic, oceanic and biogeochemical components respectively? This would probably require additional experiments such as a NorESM2-AC. We also see that the upper thermocline is much more stratified in NorESM1 than in NorESM2: why and what are the consequences? Winds and evaporation minus precipitation are the main players but we have no idea of the biases they exhibit in the non corrected model configurations. A consequence of this general concern is the discussion that is really vague and not very informative, to my opinion.**

**REPLY**: Thank you very much for raising this important point. Regarding a new experiment like **NorESM2-AC**, we agree that it is worth of performing bias-corrected simulations with NorESM2. On the other hand, as our results show (Figs. 1 and 2), NorESM2 simulations are successful in reducing the tropical Atlantic biases, which is more or less comprable with NorESM1-AC and our study is a first one to investigate and assess NorESM2 in the tropical Atlantic biogeochemical processes. Therefore, we intended to focus on NorESM2 without any corrections. In order to bias correct NorESM2, we would need more computational time and other resources, due to the more complex components, to implement the method of anomaly coupling (Toniazzo and Koseki, 2018): modifications of model's source codes, spin-up with anomaly coupling, etc. Therefore, we consider that bias -correction of NorESM2 will be more suitable for future works. We will add this brief discussion in the **Summary and Discussion**. Please see lines 466-468.

However, we agree with the reviewer that more clarification on why NorESM2 simulations can reduce the bias of the tropical Atlantic are warranted. As we described in the Section2, there are simultaneous updates of physical and biogeochemical parameterizations included in NorESM2 from NorESM1, in addition to updated atmospheric and land components. Therefore, it is not feasible to isolate which new paramerization or process improvement is responsible for the improvements in the ocean biogeochemistry. Therefore, as a first order, we examinaed the vertical structure of ocean temperature as in Fig. 2. The vertical strucuture is fundamental to investigate the model bias in the tropical Atlantic and NorESM2 experiments have better zonal gradient of the thermocline. This is an indication that ocean physical processes such as upwelling and Kelvin wave propagation (responsible for the thermocline gradient) are improved in the NorESM2. To make this implication more robust, we compare the seasonal cycle of sea surface height (SSH) among the observation and NorESM simulations as shown below in Fig.R1.

[Figure]

**Figure R1.** Hovmöller plot of sea surface height anomaly from annual mean (averaged between 3S-3N). The contour is for AVISO observation (countoru interval is 0.01m, 1993-2001) and color shadings are from different NorESM simulations.

In NorESM1-CTL, the seasonal cycle of high and low SSH in the eastern basin (20W-10E) is delayed by 1-2 months as compared to the observation. This is a common bias of ESMs generating a warm bias and a misrepresentation of the Atlantic Cold Tongue. Especially, the low SSH is a result of eastward propagation of upwelling Kelvin wave from spring to summer. The westerly trade wind is a key driver of the upwelling Kelvin wave.

In NorESM1-AC, the climatological bias correction leads to considerable improvements of the SSH seasonal cycle. Applying this methodology of bias correction, the ocean component is forced by the right surface wind and this indicates that the upwelling Kelvin wave is also realistically generated. This appears to be the primary reason for the improved marine biogeochemical processes in NorESM1-AC.

In NorESM2, the seasonal cycle of SSH remains biased (in particular, NorESM2-LM), however, the shoaling of SSH is more realistic than NorESM1-CTL, for example, the shallowest longitude is 10E (as the observation) in NorESM2 and 10W-0 in NorESM1-CTL. This indicates that the upwelling Kelvin wave propagation is represented better in NorESM2 than in NorESM1-CTL. Actuallty, the dynamics of surface wind, Kelvin wave, SSH, and SST is maintained by the Bjerknes Feedback and it is hard to quantify which components play a more important role in alleviating the bias. However, we can indicate that NorESM2 reproduces more realistic air-sea interaction than NorESM1-CTL as shown in Figs. 2 and R1 and consequently, the marine biogeochemical processes are also improved as shown in Fig.5. We have added this statement and Fig.R1 as new Fig.S3. Please see lines 201-208.

Regarding the stratification, one of possible causes might be different ocean mixing layer parameterization between NorESM1 and NorESM2. Other possobility is ocean circulation at deeper layer. As Figs. 2 and 5 show, NorESM1 has much cooler subsurface ocean and more nutrients. This indicates that upwelling of deeper ocean watermass is stronger in NorESM1. We also note that there might be other drivers for this difference, for example, the stronger AMOC in NorESM1. We already discussed this point at lines 282-284 in the original manuscript.

**3) Marine biogeochemistry is evaluated by inspecting nutrients, PP and pCO2. pCO2 is very sensitive to the dynamics (as mentioned and shown in the study) and it is thus not very surprising that any improvement in the representation of ocean dynamics has a strong impact on it. It is not a very good tracer of the ecosystem component of the biogeochemical model. PP is not observed but reconstructed from some algorithms both for chlorophyll and PP itself which are known to have significant issues (different algorithms can give very different results). I would have liked to see a comparison to chlorophyll satellite data which are much more direct and with less uncertainties.**

**REPLY**: Thank you very much for raising this important point. Yes, Chlorophyll should be an interesting variable to investigate. In the revision, we estimated Chlorophyll-a from NorESM phytoplankton data and compare it with observational data in a same manner as PP. Please note that we multiplied the phytoplankton concetration (mol P m$^{-3}$) **1000 x 122 x 12.01 / 60** to obtain the unit of mg m$^{-3}$. In addition, the modeled chrolophyll is an averaged value within the model euphotic layer (top 100m depth).

[Figure]

[Figure]

**Figure R2**. (a)-(e) Same as Fig. 3, but for 50m-mean chrolophyll-a from observation and NorESM simulations. (f) and (g) same as Fig. 5b, but for 100m-mean and 50m-mean chrolophyll-a (mg m$^{-3}$). For the NorESM simulations, phytoplankton concetration (mol P m$^{-3}$) is converted to chrolophyll-a by a factor of 1000 x 12.01 x 122 / 60.

Hovmöller plots of chlorophyll-a (Fig. R2) show quite a similar seasonal cycle as primary production (vertically-integrated in the entire water column) as given in Fig. 3: the Atlantic summer bloom in the central to eastern basin. In NorESM1 simulations, the physical bias correction improves significantly the seasonal cycle reproducing the Atlantic summer bloom in the right location and timing (Fig.R2c). Performance for chlorophyll-a is as good as for PP in NorESM1-AC. On the other hand, in NorESM2 simulations, the Atlantic summer bloom is somewhat improved from NorESM1-CTL (from June to July), but there is another high chlorophyll-a in the western basin. This can be associated with riverine flux input in NorESM2. In the observation, relatively high chlorophyll-a is also detected in the western basin from June to September (Fig.R2a). We note that there are considerable uncertainties in chlorophyll-a estimates from remote sensing, especially along the coastal regions (Gregg and Casey, 2004)

As in the scatter plot of Fig. R2f, NorESM1-AC and NorESM2 are still better than NorESM1-CTL even though NorESM2 simulations of chlorophyll-a are not as good as PP in Fig.5b. Similarly, but averaged over the top 50m dept,h mean chlorophyll-a is examined in Fig. R2g. In this depth, the performance of NorESM is as good as PP even though the magnitude of chlorophyll-a in NorESM is much larger than the observation (not shown).

According to this additional analysis, we have concluded that the NorESM's capability in reproducing PP and chlorophyll-a is roughly identical and the improvement by the physical bias correction and new generation of the model is similar to PP. We added this analysis as Supplemental Information and some descriptions in the revised manuscript. Please see lines 147-149 and 284-288.

**In summary, I think that this study needs some major revisions addressing my general concerns before it can deserve publication. A crucial point is a more thorough investigation of the features that explain the improvements obtained in the different model configurations. Finally, the model performs quite bad in terms of PP and pCO2, even in the best configurations depite what the authors state sometimes in the study. However, this is not a concern for me because ESM but also quite coarse ocean-only models tend to behave quite badly in this basin. However, I would be curious to see Chlorophyll.**

**REPLY**: Thank you very much for your constructive comment. As we answered to the previous comment, we analyzed chlorophyll-a and found that the performance is quite similar between PP and chlorophyll-a in NorESM simulations (in particular, NorESM1-CTL and NorESM1-AC), and therefore, we added the analysis of chlorophyll-a as Supplemental Information. We agree with the reviewer that other limitations like coarse resolutions also contribute to the imperfect representation of PP and sea-air $CO_2$ flux in NorESM2.

**Minor comments:**

**on the manuscript as a whole: Obviously I'm not a native English speaker, but I think the English can be improved. In addition, there are typos and formatting problems with references throughout the manuscript that should be corrected.**

**REPLY**: Thank you very much for the comment. We have read the manuscript again more carefully and corrected identified grammeratical errors and typos.

**Section 2.3: You don't explain what MPI SOM-FEM is.**

**REPLY**: Thank you very much for the comment. We have added it. Please see lines 144-146.

**Line 176: NorESM2 has a warmer subsurface and a less stratified upper thermocline (seen also on the nutrient vertical distribution), why? It relates to my general concern 2.**

**REPLY**: As we answered to the previous comment, this might be because of the difference in the AMOC between NorESM1 and NorESM2. This statement is already given in the original manuscript. Please see lines at 282-284.

**Section 3.2 and figure 3: The ACT is clearly improved, especially in NorESM2-MM but also in NorESM2-LM and is better (at least from what I can see) than NorESM1-CTL. Thus, part of the improvement is not related to the increased atmospheric resolution but to changes in the ingredients of the physical components. Any clue on what they are. Furthermore from January to June, NorESM2 is not that good and worse than NorESM1-CTL. It is significantly warmer and with two maxiam close to the African coast and 30-35W. It should be mentionend and ideally commented.**

**REPLY**: Thank you very much for this comment. As we replied to the previous comment, we have added an analysis on sea surface height (SSH) seasonal cycle in Fig. R1 (new Fig. S3 in the revised manuscript). Actually, NorESM2-LM/MM have better shoaloing features in the eastern basin of the equatorial Atlantic during summer. This indicate that NorESM2 has better ocean physics than NorESM1-CTL. In addition, NorESM2-MM has better seasonal cycle in SSH in summer. This statement is given in the revised manuscript. Please see lines at 201-208 and new Fig.S3.

Regarding the warmer SST in the western basin, we have also added some statements elaborating the potential mechanistic cause of it on it. Please see lines at 201-208.

**Lines 262-264: what are these improvements? Very vague.**

**REPLY**: Here, we mention the difference in vertical strucutre of ocean temeprature and nitrate concentration and possible causality of the difference.

**Section 3.3: Why using different types of analysis for SST and PP? Is there any reason behind the differential treatment?**

**REPLY**: It has been shown ealier (Chenillat et al. (2021) and Koseki et al. (2023)) that primary production (Chlorophyll-a) and sea-air $CO_2$ flux respond sensitively to the inter-annual variability of SST, in particular, Atlantic Niño during summer. Therefore, following their findings, we think it is crucial to first explore how well the NorESM

simulations reproduce the Atlantic Niño in terms of its intensity, peak time in summer, and location. This can be assessed in Fig. 6. Following this evaluation, it is important to investigate how the marine biogeochemical processes are influenced by the Atlantic Niño and Niña. Primary production and sea-air $CO_2$ flux are also determined by other physical and biogeochemical processes that are fluctuated directly and indirectly by the SST anomalies. Therefore, in Figs. 8 and 9, we explore the performance of key marine biogeochemical process fluctuations in response to the Atlantic Niño/Niña. In the case of primary production, upwelling of nitrate is the main drivers, consistent with Chenillat et al. (2021), while parts of sea-air $CO_2$ flux response is determined by changes in sea surface salinity (Koseki et al., 2023).

**Why not a taylor plot for PP (or even better Chlorophyll) similar to what is done with SST.**

**REPLY**: Even though the seasonal cycle of primary production improved following the better SST, it is not necessary that the Taylor plot between two variables becomes quite similar because SST is only one of functions/indicators to determine the primary production.

**Lines 380-382: the ingassing bias between 8S and 10S along Africa is quite strong in NorESM1. What is the cause of that sink. PP does not seem to be very very high at the specific location according to Figure S4. In lines 386-387, it is stated that it might be biogeochemical issues or riverine input. This is really vague and does not say anything.**

**REPLY**: Because NorESM1-AC also reduces the SST bias along the African coast, this ingassing bias in both NorESM1-CTL and NorESM1-AC might stem from marine biogeochemical process associated with the riverine flux of biogeochemical matters, which is one of main differences in biogeochemical processes between NorESM1 and NorESM2. Other possibility could be freshwater input from the Congo River. As Awo et al. (2022) showed, the SSS along the coast is influenced by Congo River plume using a high-resolution regional ocean model. There might be too much of freshwater input from the Congo River. On the other hand, our model has a coarse resolution and it is not adequate to reproduce the observed physical dynamics along the coastal region. We have added this statement. Please see lines at 420-423.

---

## Referee Report (RR1)

Dear Editor and Authors,

I have read your responses to my comments as well as I have reviewed the second version of the manuscript. I appreciate the careful consideration, thorough replies and additions to the manuscript to improve its readability. I acknowledge that physical bias correction in ESM's may help to improve strong bias, and represent a suitable mechanism to correct biogeochemical results, which are strongly linked with physical features (hydrology and circulation), and understand that those corrections justify the motivation to perform the present study.

However, as a modeler of small–scale regional domains, the results of the global models (used by the IPCC in CMIP*) and their significant (huge) biases in biogeochemical variables (even with the bias corrections made) leave me concerned about their limitations for future period studies. Of course, I understand the need to conduct studies like the present manuscript and to continue improving the ESM models.

---

## Author Response (AR2)

**Reply to the Editor's technical correction**

Dear Dr. Peter Landschützer

Thank you very much for addressing our manuscript and accepting it. We have corrected our manuscript following the editor's technical correction and upload it through the system. Again, we gratefully appreciate the editor and the two anonymous reviewers for their efforts and improving our manuscript.

**1. Please number the figures of the manuscript consecutively with the next revision (figure 1 is labelled figure 2).**

REPLY: The label of Fig.1 has been corrected.

**2. Please compare the title in the MS records with the title in the manuscript and adapt it accordingly: Model vs. Models.**

REPLY: We use "Model" and then, the title in the manuscript has been corrected.

In addition, we found some labelling errors in the manuscript for supplemental figures and those have been corrected.

Sincerely,

Shunya Koseki, on the behalf of all co-authors